# J-Shaped Relationship of Serum Uric Acid with Unfavorable Short-Term Outcomes among Patients with Acute Ischemic Stroke

**DOI:** 10.3390/biomedicines10092185

**Published:** 2022-09-04

**Authors:** Chih-Yang Liu, Cheng-Lun Hsiao, Pei-Ya Chen, Adam Tsou, I-Shiang Tzeng, Shinn-Kuang Lin

**Affiliations:** 1Stroke Center and Department of Neurology, Taipei Tzu Chi Hospital, Buddhist Tzu Chi Medical Foundation, New Taipei City 23142, Taiwan; 2School of Medicine, Tzu Chi University, Hualien 97004, Taiwan; 3Department of Research, Taipei Tzu Chi Hospital, Buddhist Tzu Chi Medical Foundation, New Taipei City 23142, Taiwan

**Keywords:** acute ischemic stroke, hyperuricemia, J-shaped correlation, mortality, uric acid, unfavorable outcomes

## Abstract

(1) Background: The role of uric acid in stroke outcomes remains inconclusive. (2) Methods: We retrospectively enrolled 3370 patients with acute ischemic stroke. (3) Results: Uric acid level was higher in men than in women. Univariate analyses revealed that the rates of hyperuricemia were higher in all patients and in women for unfavorable outcomes. For death, the hyperuricemia rates were higher in all patients including men and women, and the uric acid levels were also higher in all patients and in women. A J-shaped curve was observed between uric acid and the discharge-modified Rankin Scale score. Patients within Quartiles 1 (<4.1 mg/dL) and 4 (>6.5 mg/dL) of uric acid had higher rates of unfavorable outcomes and death than patients within Quartiles 2 (4.1–5.1 mg/dL) and 3 (5.1–6.2 mg/dL). Multivariable analyses for unfavorable outcomes revealed that Quartile 1 of uric acid was a significant factor in all patients and in men. In men, a significant factor for death was being in Quartile 1 of uric acid. In women, higher levels of uric acid or hyperuricemia (>6.6 mg/dL) were significant factors for death. (4) Conclusions: Lower uric acid levels are a predictor for unfavorable outcomes and death in men, and higher uric acid levels are a predictor for death in women.

## 1. Introduction

Significant stroke comorbidities are associated with conventional risk factors for vascular diseases, including old age, hypertension, diabetes mellitus (DM), dyslipidemia, and heart disease. Evidence has shown that healthy blood pressure, serum sugar and lipid reduce stroke risk and lead to better functional outcomes after a stroke [1]. Unlike conventional risk factors, the role of uric acid in stroke risk and stroke outcomes remains inconclusive.

Uric acid is an end-metabolic product of purine that exists in the form of uric acid salt with high solubility in organisms. Uric acid has an antioxidant effect to protect the body but also a proinflammatory effect to stimulate atherosclerosis. Excessive blood uric acid can induce gout or kidney stone. An absence of gout flares does not necessary imply the absence of uric acid-related tissue damage. Elevated uric acid is associated with hypertension, obesity, DM, increasing risk of stroke, and poorer stroke outcomes [2,3]. Hyperuricemia has been shown as a risk factor of stroke [4]. While some studies have emphasized the beneficial effect of higher uric acid levels on better stroke outcomes due to its antioxidant properties and its 10-fold serum concentration being higher than other antioxidants in the human body [5,6]. Other studies have failed to identify any correlation between uric acid and stroke prognosis. Furthermore, the normal range of serum uric acid levels varies in men and women. These diverse natures of uric acid result in research or statistical barriers. In this study, we aimed to investigate uric acid level, quartile distribution of uric acid, and hyperuricemia, and their correlations with clinical features and short-term outcomes in patients with acute ischemic stroke (AIS).

## 2. Materials and Methods

### 2.1. Study Population and Data Collection

To control for differences in stroke care, we registered patients who were admitted to the ward for acute stroke. We retrospectively reviewed the records of all registered inpatients who presented with AIS from January 2011 to December 2021. The diagnosis of AIS was confirmed by clinical presentation and proof of either an ischemic lesion or absence of a corresponding intracranial lesion other than infarction by using brain-computed tomography or magnetic resonance imaging. The following information was collected: age, sex, body mass index (BMI), and stroke risk factors, including history of hypertension, DM, dyslipidemia, heart disease, prior stroke, smoking status, alcohol consumption, cancer, and uremia. We collected laboratory data on arrival at the emergent department, including a full blood count with white blood cell differentials and platelet, glucose, and creatinine. We obtained fasting cholesterol, triglyceride, and uric acid levels within 24 h of admission. We also collected in-hospital complications and length of hospital stays.

### 2.2. Statement of Ethics

The study was conducted in accordance with the Declaration of Helsinki. Ethical approval for this study was provided by the Institutional Review Board of Taipei Tzu Chi Hospital, New Taipei City (approval no. 11-X-111). Informed written consent was waived because this study conducted retrospective data analysis. All data collected and analyzed were derived from clinical records without any intervention or influence on clinical treatment. Only clinical observation data were used for publication, and personal information was not disclosed to any other third party without the patient’s consent to protect patient privacy and rights.

### 2.3. Stroke Severity, Classification, and Clinical Features

Stroke severity was assessed on admission according to the National Institutes of Health Stroke Scale (NIHSS). We classified the etiology of ischemic stroke according to the Trial of ORG 10172 in Acute Stroke Treatment (TOAST) categories, namely large-artery atherosclerosis, small-vessel occlusion, cardioembolism, other determined etiology, and undetermined etiology [7]. Pneumonia, urinary tract infection, gastrointestinal bleeding, and seizure were registered as in-hospital stroke complications. The normal values of serum uric acid in the indexed hospital range were 4.4 to 7.6 mg/dL for men and from 2.3 to 6.6 mg/dL for women. A uric acid level of >7.6 mg/dL in men and of >6.6 mg/dL in women indicated hyperuricemia. At discharge, functional outcomes were evaluated using the NIHSS, the Barthel index, and the modified Rankin Scale (mRS). An mRS score of >2 indicated an unfavorable outcome (including death).

### 2.4. Statistical Analysis

Given the relatively large size of patient numbers, continuous variables are presented as the mean ± standard deviation, and group comparisons of continuous variables were performed using two-sample *t* tests or analysis of variance, as appropriate. A chi-square test or Fisher’s exact test were used for categorical comparisons. We analyzed correlations of continuous variables using linear regression or least squares regression test. Patients were further stratified into four groups according to the quartile distribution of serum uric acid levels. Significant factors in univariate analyses were added to a multiple logistic regression model to identify the significant factors associated with unfavorable outcomes or death. Significance was indicated if *p* < 0.05. All statistical analyses were performed using SPSS (version 24; SPSS, Chicago, IL, USA). Scatter plot diagrams were constructed using MedCalc version 18 (MedCalc Software, Mariakerke, Belgium).

## 3. Results

During the study period, 3370 patients with AIS were enrolled, including 1925 men and 1445 women, with an average age of 70.6 years. Table 1 summarizes the clinical features, laboratory data, and outcomes. The average uric acid value in all patients was 5.3 mg/dL. Hyperuricemia, unfavorable outcomes, and death were observed in 12%, 49%, and 4% of all patients, respectively. Women were older, had higher platelet and cholesterol levels, higher heart disease rates, more history of cancer, and higher rates of hyperuricemia. Men had higher BMI, hemoglobin levels, creatinine levels, triglyceride and uric acid levels, and higher rates of prior stroke, smoking, and alcohol consumption. Women had a higher rate of in-hospital complications and longer lengths of hospital stays. Both NIHSS scores on admission and at discharge were higher in women. The functional outcomes were also worse in women regarding lower Barthel index score, higher mRS score, and a higher rate of mRS score > 2 at discharge. No difference in mortality was observed between men and women.

Table 2 summarizes the univariate analyses of factors associated with unfavorable outcomes. Patients with unfavorable outcomes were older, had higher levels of NIHSS score on admission, white blood cell counts, glucose, and creatinine, and higher rates of hypertension, DM, heart disease, prior stroke, history of cancer, and uremia, but lower levels of hemoglobin, platelet, cholesterol, and triglyceride, and lower rates of dyslipidemia, current smoker, and alcohol consumption. We observed no difference in uric acid levels. However, the rate of hyperuricemia was higher in patients with unfavorable outcomes. Most factors associated with unfavorable outcomes remained the same after dividing the patients into men and women except the rate of alcohol consumption, which was not significant in either sex. The levels of white blood cells and glucose and the rates of hypertension, history of cancer, and hyperuricemia were not significantly different in male patients. Women with unfavorable outcomes tended to have higher uric levels (*p* = 0.051).

Table 3 summarizes the univariate analyses of factors associated with death. Patients who met with fatal outcomes were older, had higher NIHSS scores on admission, had higher white blood cell count, glucose, creatinine, uric acid, and higher rates of heart disease, history of cancer, uremia, and hyperuricemia, but lower levels of hemoglobin, cholesterol, and triglyceride. Older age, higher levels of NIHSS score on admission and white blood cells, and higher heart disease and hyperuricemia rates remained significant in gender-stratified subgroups. However, hemoglobin levels were still significantly lower in men, and glucose, creatinine, and uric acid levels were still significantly higher in women. Notably, platelet levels were lower in men, and the rate of DM was higher in women.

The additional analysis of the association of the uric acid levels with unfavorable outcomes or death in 400 patients with hyperuricemia by using the Mann–Whitney U test showed that only women with fatal outcome had significant higher level of uric acid than women without fatal outcomes (8.1 (7.3–9.3) mg/dL vs. 7.6 (7.1–8.4) mg/dL, *p* = 0.039). No differences of uric acid levels were observed among all patients, men, and women with and without unfavorable, and among all patients and men with and without death.

We explored the role of uric acid due to a discrepancy between the measured value of uric acid and categorized hyperuricemia. Table 4 shows the distributions of uric acid levels and discharge mRS in groups based on the TOAST classification. The level of uric acid and the rate of hyperuricemia were highest in patients with cardioembolism and lowest in patients with other determined etiology. Patients with large-artery atherosclerosis and cardioembolism had the highest levels of discharge mRS score and the highest rate of unfavorable outcomes. In addition, patients with cardioembolism had the highest rate of mortality. Uric acid was positively and linearly correlated with BMI, hemoglobin, white blood cells, creatinine, cholesterol, and triglyceride, and negatively and linearly correlated with glucose (Table 5). Uric acid had a trend of positive linear correlation with admission NIHSS score but had no linear correlation with discharge NIHSS, Barthel index, and mRS scores. Hyperuricemia was observed more in patients with hypertension, dyslipidemia, heart disease, in-hospital complications, and unfavorable and fatal outcomes. When the sample was segmented by gender, uric acid was linearly correlated with age, discharge NIHSS score, and mRS score in women. Men and women with fatal outcomes had higher rates of hyperuricemia, but only women with unfavorable outcomes had higher rates of hyperuricemia.

Given that the positive linear correlation between uric acid and discharge mRS score was observed only in women, we conducted least squares regression analyses between uric acid and discharge mRS score to identify the best fit of the correlating curve. A J-shaped pattern of a curve of the relationship between uric acid and discharge mRS score with a turning point present at a point where the uric acid concentration is 5–6 mg/dL was observed in all 3370 patients (Figure 1A). The J-shaped correlations were still present in gender-stratified subgroups (Figure 1B,C). These J-shaped patterns of curves revealed that both lower and higher uric acid levels were related to higher discharge mRS scores.

We further stratified all patients into four groups according to the quartile distribution of the uric acid, namely Quartile 1 (<25 percentile; uric acid < 4.1 mg/dL), Quartile 2 (25–50 percentile; uric acid concentration = 4.1–5.1 mg/dL), Quartile 3 (51–75 percentile; uric acid concentration = 5.2–6.2 mg/dL), and Quartile 4 (>75 percentile; uric acid > 6.2 mg/dL). We observed no age differences among the four quartiles (Table 6). Women were more prevalent in Quartile 1 subgroup, whereas men were more prevalent in Quartile 4 subgroup. Unfavorable outcomes were similar in patients in Quartiles 2 and 3. Patients categorized in Quartiles 1 and 4 had higher NIHSS scores on admission and higher rates of unfavorable outcomes and death than patients in the second and third quartiles. Among those, patients in the first quartile had the highest rate of unfavorable outcomes and patients in Quartile 4 had the highest rate of death. All the results remained the same after dividing the patients into men and women with the quartile distribution derived from men or women separately, except the number of patients categorized in each quartile and higher initial NIHSS score in men within Quartile 4 subgroup (Appendix A). Based on these findings, we performed two linear regressions of the relationship between uric acid and discharge mRS score, one for patients in the Quartiles 1 and 2 subgroups (Group A), and another for patients in the Quartiles 3 and 4 subgroups (Group B). In patients in Groups A and B, uric acid level had a negative and positive correlation with discharge mRS score, respectively (Figure 2).

These results differed between men and women. Among men, the differences in admission NIHSS scores among the four quartiles were nonsignificant, and the rates of unfavorable outcomes and death were highest in Quartile 1. Among women, patients in Quartile 4 were older and had the highest NIHSS scores on admission and the highest R2 has been changed to r2rates of unfavorable outcomes and death. Figure 3 illustrates the distribution patterns of admission NIHSS score, percentages of unfavorable outcomes (discharge mRS score > 2), and death among patients for the four quartiles of uric acid levels. The admission NIHSS scores were higher in all patients with uric acid levels within Quartiles 1 and 4. Unfavorable outcomes and death rates were highest among men for Quartile 1 of uric acid and were highest among women for Quartile 4 of uric acid.

Given that both Quartiles 1 and 4 of uric acid were correlated with unfavorable outcomes and that the rate of hyperuricemia was higher in patients with unfavorable outcomes, we included uric acid quartile derived from all patients as a variable in a Model I and the presence of hyperuricemia as a variable in a Model II in a multiple logistic regression to identify factors influencing unfavorable outcomes. In addition, we included uric acid level as a variable in a Model III to identify the factors associated with death. In Model I, Quartiles 2 and 3 of uric acid were taken as reference percentiles. Table 7 presents the multiple regression results for unfavorable outcomes. BMI was not included in the multiple regression model because it had too many missing data points (BMI data were available for 76.4% of patients). For all 3370 patients in Model I, the significant factors for unfavorable outcomes were age, admission NIHSS score, white blood cells, glucose, cholesterol, prior stroke, history of cancer, and Quartile 1 of uric acid. Quartile 4 of uric acid was not a significant factor. Hyperuricemia was not a significant factor for unfavorable outcomes in Model II. Using the same analysis with Model I, we conducted the multiple logistic regression analysis in men and women separately by selecting corresponding significant factors observed from univariate analyses in Table 2. Compared with the significant factors in all 3370 patients, white blood cells, glucose, and history of cancer were not considered significant factors in men. Quartile 1 of uric acid was a stronger predictor for unfavorable outcomes in men. For women, Quartiles 1 and 4 of uric acid and hyperuricemia were not significant factors for unfavorable outcomes.

Table 8 presents the results of the multivariable analyses for death by selecting corresponding significant factors from univariate analyses in Table 3. The multiple logistic regression results for all 3370 patients revealed that significant factors for death were age, admission NIHSS score, white blood cells, heart disease, and history of cancer. No uric acid-related variables were statistically significant in Models I to III. For men, significant factors for death were age, admission NIHSS score, white blood cells, and being in Quartile 1 of uric acid (Model I). Among women, the significant factors for death were admission NIHSS score, white blood cell count, creatinine level, heart disease, hyperuricemia (Model II), and higher uric acid level (Model III). All the results remained the same after dividing the patients into men and women with the quartile distribution derived from men or women separately in Model I (Appendix A). To simplify the clinical application, we prefer to use the same quartile distribution derived from all patients for both men and women.

## 4. Discussion

The defined normal range of uric acid in men is higher than that in women, and uric acid’s influence on the outcomes differed between men and women. We observed a J-shaped curve pattern between the uric acid and the discharge mRS scores in all patients and both male and female subgroups. Compared with men, women had lower uric acid levels but a higher incidence of hyperuricemia. Unfavorable outcomes were correlated with a lower uric acid level within Quartile 1 in all patients and male patients. No correlation between uric acid with unfavorable outcomes was observed in female patients. We observed no correlation between death and uric acid across all patients. However, death was correlated with lower uric acid levels within Quartile 1 in male patients, and death was correlated with higher levels of uric acid and hyperuricemia in female patients.

Scholars have argued that hyperuricemia and elevated serum uric acid are associated with increased stroke incidence and mortality risks [4,8,9,10]. However, several studies have demonstrated that uric acid and vascular events have a complex relationship, with different presentations between men and women. The PIUMA study demonstrated a J-shaped relationship between uric acid and cardiovascular events in both genders [11]. A J-shaped relationship between uric acid and the risk of ischemic stroke was also observed by Hu et al. [12]. A j-curve relationship between uric acid and risk of cardiometabolic diseases was observed only in women by Kuwabara et al. [13]. A dose–response meta-analysis conducted by Qiao et al. revealed a J-shaped trend between ascending serum uric acid levels and a higher risk of suffering from stroke, both for ischemic and hemorrhagic strokes, especially in females [14]. They found that the association between serum uric acid levels and risk of stroke became statistically significant when uric acid reached 5.35 mg/dL.

Contrary to the previously mentioned relationship between higher uric acid levels and poor outcomes, Tang et al. found that lower uric acid levels are associated with poor outcomes [15]. However, higher uric acid levels have been associated with better stroke outcomes in some studies, and in patients with thrombolysis [5,16,17,18,19]. Furthermore, several studies have reported a U-shaped relationship between uric acid and poor stroke outcomes (including mortality) [20,21,22,23]. Recently, a J-shaped pattern in the relationship between uric acid and mortality was observed in patients with DM [24]. A retrospective study conducted by Browne et al. in a large cohort of patients within a health-care system found a J-shaped pattern in the relationship between uric acid level and all-cause mortality in women and a U-shaped pattern in men [25]. J-shaped relationships between uric acid and stroke outcome are seldomly reported. If the upward curves on both sides of the line with high and low uric acid levels are not precisely demonstrated, the J-shaped curve might be interpreted as a “U-shaped” curve, and vice versa. In this study, we establish a J-shaped relationship between the uric acid level and the discharge mRS score in all patients and both male and female patients. Therefore, abnormally low or high uric acid levels are associated with unfavorable outcomes.

Uric acid is an end catabolite of purine nucleotide metabolism. Purines are normally produced in the body, but they can also be found in some foods and drinks. Uric acid is either produced from endogenous tissue catabolism from RNA and DNA bases or exogenous sources of high purine intake. The definitions of hyperuricemia are variable and have been defined as >7.7 mg/dL in men and >6.6 mg/dL in women, or >7.0 mg/dL in men and >6 mg/dL in women [26,27,28,29]. Bardin et al. proposed several definitions of hyperuricemia, including statistical and physicochemical definitions of hyperuricemia [26]. A possible explanation for the difference between genders is the interaction between estrogen on the enhanced uric acid excretion in females, along with the suppression of uric acid transporter 1 in the proximal tubule [30]. Uric acid level increases significantly with age in women, particularly after menopause [31]. However, uric acid levels are not significantly correlated with age in men. In the present study, we also found similar results in Table 5. We observed no linear correlation between uric acid and age among all patients. However, uric acid level exhibited a positive linear correlation with age among women but not men. Given that the kidney excretes approximately two-thirds of serum uric acid, the uric acid level also exhibited a positive linear correlation with the creatinine level (Table 5). Uric acid provides an antioxidant defense against oxidant- and radical-caused damage in humans [32]. It provides approximately 60% of the free-radical scavenging capacity of the blood, with its concentration increasing significantly during acute stroke [29]. During an acute stroke, the inflammatory response and oxidative stress may cause extensive damage to the surrounding brain tissue [33,34]. Uric acid has the useful ability to clear out peroxynitrite, nitric oxide, and hydroxyl radicals. It can therefore protect nerves from oxidative damage during an acute stroke and prevent worse outcomes [14,18,19]. However, elevated uric acid was also reported to promote atherosclerotic progression by increasing the production of free radicals and facilitating low-density lipoprotein cholesterol oxidation [35]; to increase the level of inflammatory cytokines; to increase platelet aggregation and thrombus formation [36]; and to be associated with carotid intima-media thickness [37], proximal extracranial artery stenosis [38], and intracranial arterial stenosis [39]. Playing various roles in these mechanisms, uric acid functions as a double-edged sword. The aggravated inflammatory response due to too high uric acid levels and the reduced antioxidant neuroprotective effect from overly low uric acid levels can increase the likelihood of unfavorable outcomes in patients with ischemic stroke. Such influences differ by sex, owing to differences in uric acid metabolism.

The neuroprotective antioxidant effect of uric acid has been documented by several studies in which patients with higher levels of uric acid during AIS exhibited a better outcome [5,6,14,18,19], particular in men [16,17], and patients with lower levels of uric acid had worse outcome [15,40]. Similar observation has been reported that patients with multiple sclerosis, neuromyelitis optica, and Parkinson’s disease showed lower uric acid levels when compared with healthy controls [41,42]. Compared to men, women have lower endogenous levels of uric acid and are therefore at greater risk of exposure to unopposed toxic-free radicals. In the URICO-ICTUS trial, administration of uric acid in patients treated with alteplase during AIS showed an overall nonsignificant 6% increment of good outcome [43]. Their subgroup analysis showed that the uric acid therapy doubled the placebo effect to attain a better outcome in women but not in men. However, hyperuricemia-associated risk factors for stroke, such as hypertension, diabetes mellitus, and metabolic syndrome, have been reported to occur more in women than men and might offset the neuroprotective effects of uric acid [6,17].

Both the J-shaped pattern of and the sex differences in the effect of uric acid on stroke outcomes complicate the application of statistical methods. Therefore, we interpreted the statistical results separately for men and women. The quartile distributions of uric acid in Figure 3 indicate that the risks of unfavorable outcomes and death were higher in men for Quartile 1 of uric acid but were higher in women for Quartile 4 of uric acid. Such effects persisted in multivariable analyses where various risk factors were included. In summary, for all patients, uric acid level in Quartile 1 (<4.1 mg/dL) is a significant predictor of unfavorable outcomes (*p* = 0.013) and a uric acid level in Quartile 4 (>6.2 mg/dL) is a significant predictor of fatality (*p* = 0.088). For men, a uric acid level in Quartile (<4.4 mg/dL) is a significant predictor of both unfavorable (*p* = 0.005) and fatal (*p* = 0.002) outcomes. Among female patients, none of these quartile levels of uric acid are significant predictors; however, higher uric acid levels (*p* = 0.029) and the presence of hyperuricemia (>6.6 mg/dL; *p* = 0.047) are predictors of fatal outcomes. Despite the J-shaped relationship patterns between uric acid level and discharge mRS score in men, women, and all patients, we can conclude generally that the risk is higher in men with lower uric acid levels and higher in women with higher uric acid levels. The clinical significance of this study shows that uric acid should be regarded as one of the risk factors for stroke. The detection of uric acid value either during daily outpatient practice or during acute stroke is crucial and is associated with stroke prognosis. For men, we recommend to avoid too low levels of uric acid below 4.1 mg/dL, and for women, the uric acid level needs to be controlled below 6.6 mg/dL.

This study has several limitations. First, BMI data were only available in 76.4% (2575/3370) of patients; therefore, we did not include it in the multivariable analysis. BMI can be difficult to measure in patients presenting with severe neurological deficits. Therefore, we could not compare the correlation of uric acid and BMI with outcomes. A study demonstrated that low uric acid and low BMI jointly influence the probability of unfavorable outcomes in patients who had ischemic stroke [16]. A second limitation is that the values in the definition of hyperuricemia (>7.6 mg/dL in men and >6.6 mg/dL in women) in this study is higher than those in commonly used criteria (>7.0 mg/dL in men and >6.0 mg/dL in women). Nevertheless, our criteria are similar to those used in studies conducted by Chang et al. [44] and Chien et al. [45] among the Taiwanese population. We analyzed uric acid with three models (Models I to III) that each included uric acid quartile, the hyperuricemia, or uric acid level as variables. Compared with hyperuricemia, uric acid level and uric acid quartile are more valuable predictors of outcomes. Finally, this is a retrospective study with data limited to the discharge outcome. Therefore, we could not determine an intermediate outcome at 3 months nor conduct a long-term follow-up.

## 5. Conclusions

The J-shaped relationship between uric acid and the discharge mRS score implies that uric acid has both positive and negative consequences on stroke outcomes. Overly low or high uric acid levels are associated with unfavorable outcomes. A low uric acid level within Quartile 1 (<4.1 mg/dL) predicts unfavorable outcomes and death in men. A high uric acid level with hyperuricemia (>6.6 mg/dL) predicts death in women.

## Figures and Tables

**Figure 1 biomedicines-10-02185-f001:**
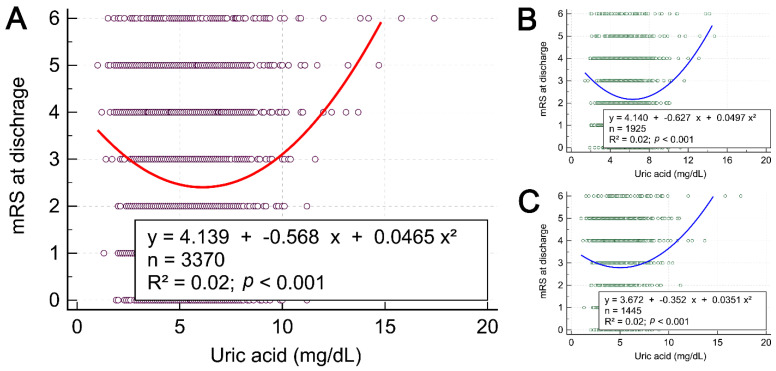
Least squares regression analyses show a J-shaped curve between uric acid and discharge modified Rankin Scale scores in all patients (**A**), in men (**B**), and in women (**C**). mRS, modified Rankin Scale.

**Figure 2 biomedicines-10-02185-f002:**
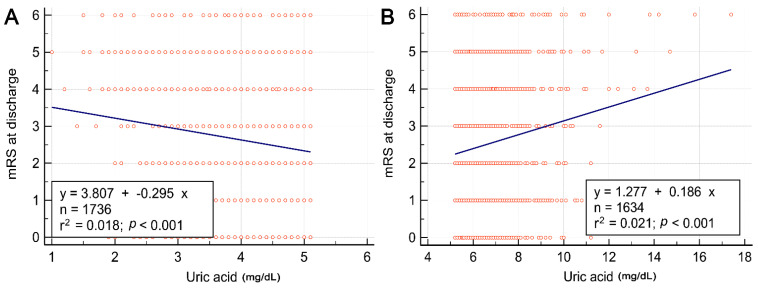
Linear correlation between uric acid and modified Rankin Scale score in patients within Quartiles 1 and 2 of uric acid (**A**) and in patients within Quartiles 3 and 4 of uric acid (**B**). mRS, modified Rankin Scale.

**Figure 3 biomedicines-10-02185-f003:**
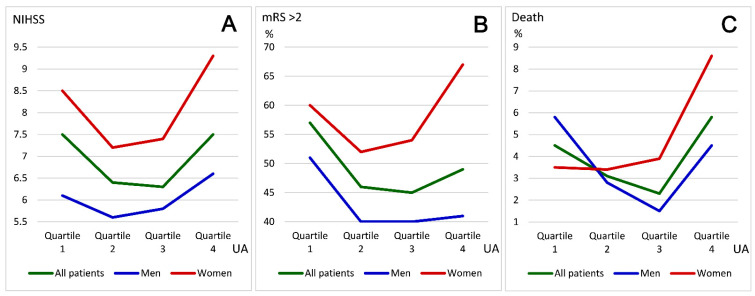
(**A**) Distributions of the average NIHSS scores according to the quartiles of uric acid in all patients, men, and women. (**B**) Distributions of the percentage of unfavorable outcomes (discharge modified Rankin Scale score > 2) according to the quartiles of uric acid in all patients, men, and women. (**C**) Distributions of the percentage of death according to the quartiles of uric acid in all patients, men, and women. The percentages of unfavorable outcomes and death were highest in men within Quartile 1 of uric acid, and were highest in women within Quartile 4 of uric acid. Ranges of uric acid levels in Quartiles 1 to 4 are <4.1 mg/dL, 4.1–5.1 mg/dL, 5.1–6.2 mg/dL, and >6.2 mg/dL, respectively; mRS, modified Rankin Scale.

**Table 1 biomedicines-10-02185-t001:** Baseline data of clinical features in 3370 patients with acute ischemic stroke.

Characteristics	All Patients (*n* = 3370)	Gender		
		Men (*n* = 1925)	Women (*n* = 1445)	*p*-Value
Age (years)	70.6 ± 13.6	68.1 ± 13.4	74.0 ± 13.0	<0.001
Body mass index	25.0 ± 4.2	25.3 ± 4.0	24.5 ± 0.4	<0.001
Hemoglobin (g/dL)	13.6 ± 2.1	14.3 ± 1.9	12.8 ± 1.9	<0.001
Platelet (×10^9^/L)	219 ± 76	212 ± 73	227 ± 79	<0.001
White blood cells (×10^3^/mL)	7.98 ± 2.77	8.06 ± 2.71	7.87 ± 2.84	0.057
Glucose (mg/dL)	163 ± 79	162 ± 76	164 ± 83	0.409
Creatinine (mg/dL)	1.3 ± 1.1	1.4 ± 1.1	1.2 ± 1.1	<0.001
Cholesterol (mg/dL)	170 ± 43	166 ± 41	174 ± 44	<0.001
LDL-cholesterol (mg/dL)	108 ± 35	107 ± 35	109 ± 36	0.187
Triglyceride (mg/dL)	124 ± 95	128 ± 105	119 ± 79	0.007
Uric acid (mg/dL)	5.3 ± 1.7	5.5 ± 1.7	4.9 ± 1.7	<0.001
Hypertension	2363 (70)	1329 (69)	1034 (72)	0.114
Diabetes mellitus	1157 (34)	654 (34)	203 (35)	0.613
Dyslipidemia	718 (21)	403 (21)	315 (22)	0.545
Heart disease	1085 (32)	573 (30)	512 (35)	<0.001
Prior stroke	769 (23)	469 (24)	310 (21)	0.047
Current smoker	717 (21)	662 (34)	65 (4)	<0.001
Alcohol consumption	227 (7)	220 (11)	7 (0.5)	<0.001
History of cancer	220 (7)	106 (6)	114 (8)	0.006
Uremia	67 (2)	34 (2)	33 (2)	0.319
Hyperuricemia	400 (12)	191 (10)	209 (14)	<0.001
In-hospital complications	373 (11)	173 (9)	200 (14)	<0.001
Length of stay (days)	14.1 ± 12.6	13.4 ± 12.6	15.0 ± 12.6	<0.001
NIHSS score on admission	6.9 ± 7.2	6.0 ± 6.5	8.1 ± 7.8	<0.001
NIHSS score at discharge	6.5 ± 9.4	5.8 ± 8.8	7.7 ± 10.1	<0.001
Barthel index score at discharge	67 ± 64	72 ± 34	59 ± 37	<0.001
mRS score at discharge	2.6 ± 1.7	2.3 ± 1.7	2.9 ± 1.8	<0.001
mRS >2 at discharge	1658 (49)	818 (42)	840 (58)	<0.001
Death at discharge	133 (3.9)	68 (3.5)	65 (4.5)	0.180

Data are expressed as mean ± standard deviation or n (%); mRS, modified Rankin Scale; NIHSS, National Institutes of Health Stroke Scale; TOAST, Trial of ORG 10172 in Acute Stroke Treatment.

**Table 2 biomedicines-10-02185-t002:** Univariate analysis of factors related to unfavorable outcomes (modified Rankin Scale score > 2) in 3370 patients with acute ischemic stroke.

Characteristics	All Patients			Gender					
				Men (n = 1925)			Women (n = 1445)		
	No (*n* = 1712)	Yes (*n* = 1658)	*p*-Value	No (*n* = 1107)	Yes (*n* = 818)	*p*-Value	No (*n* = 605)	Yes (*n* = 840)	*p*-Value
Age (years)	66.2 ± 12.9	75.2 ± 12.8	<0.001	64.8 ± 12.5	72.5 ± 13.5	<0.001	68.6 ± 13.1	77.9 ± 11.5	<0.001
Admission NIHSS score	3.1 ± 3.1	10.9 ± 7.9	<0.001	3.0 ± 2.9	10.1 ± 7.6	<0.001	3.1 ± 3.4	11.7 ± 8.2	<0.001
Hemoglobin (g/dL)	14.0 ± 1.9	13.2 ± 2.1	<0.001	14.5 ± 1.8	13.8 ± 2.1	<0.001	13.0 ± 1.8	12.7 ± 1.9	<0.001
Platelet (×10^9^/L)	222 ± 70	215 ± 82	0.003	215 ± 65	208 ± 83	0.030	235 ± 77	221 ± 80	<0.001
White blood cells (×10^3^/mL)	7.78 ± 2.50	8.19 ± 3.03	<0.001	7.9 ± 2.5	8.2 ± 2.9	0.096	7.43 ± 2.40	8.19 ± 3.07	<0.001
Glucose (mg/dL)	159 ± 76	166 ± 82	0.012	162 ± 75	162 ± 77	0.904	155 ± 77	170 ± 86	<0.001
Creatinine (mg/dL)	1.2 ± 1.0	1.4 ± 1.3	<0.001	1.3 ± 0.9	1.5 ± 1.3	0.003	1.0 ± 1.0	1.3 ± 1.2	<0.001
Cholesterol (mg/dL)	172 ± 42	167 ± 44	<0.001	168 ± 39	163 ± 44	0.005	179 ± 45	171 ± 43	<0.001
Triglyceride (mg/dL)	133 ± 91	115 ± 98	<0.001	137 ± 97	115 ± 114	<0.001	126 ± 78	114 ± 80	0.006
Uric acid (mg/dL)	5.3 ± 1.5	5.3 ± 1.9	0.286	5.6 ± 1.5	5.5 ± 1.8	0.241	4.9 ± 1.5	5.0 ± 1.9	0.051
Female gender	605 (35)	840 (51)	<0.001	-	-	-	-	-	-
Hypertension	1162 (68)	1201 (72)	0.004	754 (68)	575 (70)	0.306	408 (67)	626 (75)	0.003
Diabetes mellitus	537 (31)	620 (37)	<0.001	354 (32)	300 (37)	0.032	183 (30)	320 (38)	0.002
Dyslipidemia	416 (24)	302 (18)	<0.001	256 (23)	147 (18)	0.006	160 (26)	155 (18)	<0.001
Heart disease	428 (25)	657 (40)	<0.001	271 (24)	302 (37)	<0.001	157 (26)	355 (42)	<0.001
Prior stroke	298 (17)	481 (29)	<0.001	211 (19)	258 (32)	<0.001	87 (14)	223 (27)	<0.001
Current smoker	465 (27)	262 (16)	<0.001	427 (39)	235 (29)	<0.001	38 (6)	27 (3)	0.006
Alcohol consumption	139 (8)	88 (5)	0.001	135 (12)	85 (10)	0.246	4 (0.7)	3 (0.4)	0.461
History of cancer	91 (5)	129 (8)	0.004	55 (5)	51 (6)	0.266	36 (6)	78 (9)	0.023
Uremia	21 (1)	46 (3)	0.001	11 (1)	23 (3)	0.004	10 (2)	23 (3)	0.212
Hyperuricemia	166 (10)	234 (14)	<0.001	101 (9)	90 (11)	0.189	65 (11)	144 (17)	<0.001

Data are expressed as mean ± standard deviation or n (%); CI, confidence interval; NIHSS, National Institutes of Health Stroke Scale; OR, odds ratio.

**Table 3 biomedicines-10-02185-t003:** Univariate analysis of factors related to death in 3370 patients with acute ischemic stroke.

Characteristics	All patients			Gender					
				Men (n = 1925)			Women (n = 1445)		
	No (*n* = 3237)	Yes (*n* = 133)	*p*-Value	No (*n* = 1857)	Yes (*n* = 68)	*p*-Value	No (*n* = 1380)	Yes (*n* = 65)	*p*-Value
Age (years)	70.3 ± 13.5	78.1 ± 13.5	<0.001	67.8 ± 13.4	74.8 ± 14.7	<0.001	73.7 ± 12.9	81.4 ± 11.3	<0.001
Admission NIHSS score	6.4 ± 6.6	19.4 ± 9.1	<0.001	5.6 ± 5.8	18.6 ± 9.8	<0.001	7.5 ± 7.3	20.2 ± 8.2	<0.001
Hemoglobin (g/dL)	13.7 ± 2.0	12.9 ± 2.4	<0.001	14.2 ± 1.9	13.4 ± 2.4	<0.001	12.8 ± 1.9	12.5 ± 2.2	0.139
Platelet (×10^9^/L)	218 ± 73	221 ± 123	0.697	212 ± 69	230 ± 150	0.049	227 ± 78	212 ± 87	0.127
White blood cells (×10^3^/mL)	7.91 ± 2.70	9.59 ± 3.69	<0.001	8.00 ± 2.66	9.64 ± 3.56	<0.001	7.79 ± 2.76	9.53 ± 3.85	<0.001
Glucose (mg/dL)	162 ± 79	182 ± 83	0.004	161 ± 76	172 ± 67	0.248	163 ± 82	192 ± 96	0.005
Creatinine (mg/dL)	1.3 ± 1.1	1.7 ± 1.3	<0.001	1.4 ± 1.1	1.6 ± 1.0	0.127	1.2 ± 1.1	1.7 ± 1.6	<0.001
Cholesterol (mg/dL)	190 ± 42	164 ± 54	0.266	166 ± 40	165 ± 60	0.871	175 ± 44	164 ± 47	0.051
Triglyceride (mg/dL)	125 ± 94	110 ± 107	0.092	129 ± 106	102 ± 67	0.048	119 ± 76	118 ± 135	0.922
Uric acid (mg/dL)	5.3 ± 1.7	5.8 ± 2.7	<0.001	5.5 ± 1.6	5.6 ± 2.5	0.574	4.9 ± 1.7	5.9 ± 2.9	<0.001
Female gender	1380 (43)	65 (49)	0.180	-	-	-	-	-	-
Hypertension	2276 (70)	87 (65)	0.246	1286 (69)	43 (63)	0.288	990 (72)	44 (68)	0.483
Diabetes mellitus	1106 (34)	51 (38)	0.351	634 (34)	20 (29)	0.514	472 (34)	31 (48)	0.032
Dyslipidemia	698 (22)	20 (15)	0.083	393 (21)	10 (15)	0.227	305 (22)	10 (15)	0.222
Heart disease	1003 (31)	82 (62)	<0.001	537 (29)	36 (53)	<0.001	466 (34)	46 (71)	<0.001
Prior stroke	755 (23)	24 (18)	0.173	455 (25)	14 (21)	0.565	300 (22)	10 (15)	0.279
Current smoker	706 (22)	21 (16)	0.107	642 (35)	20 (29)	0.436	64 (5)	1 (2)	0.361
Alcohol consumption	222 (7)	5 (4)	0.214	215 (12)	5 (7)	0.337	7 (0.5)	0 (0)	0.999
History of cancer	204 (6)	16 (12)	0.018	100 (5)	6 (9)	0.269	104 (8)	10 (15)	0.032
Uremia	60 (2)	7 (5)	0.015	31 (2)	3 (4)	0.116	29 (2)	4 (6)	0.057
Hyperuricemia	366 (11)	34 (26)	<0.001	179 (10)	12 (18)	0.038	187 (14)	22 (34)	<0.001

Data are expressed as mean ± standard deviation or n (%); CI, confidence interval; NIHSS, National Institutes of Health Stroke Scale; OR, odds ratio.

**Table 4 biomedicines-10-02185-t004:** Distributions of serum uric acid levels and outcomes by TOAST classification in 3370 patients with acute ischemic stroke.

TOAST Classification	Uric Acid	Hyperuricemia	mRS Score	mRS Score > 2	Death
Small artery occlusion (n = 1485)	5.2 ± 1.5	134 (9)	1.9 ± 1.4	497 (33)	2 (0.1)
Large artery atherosclerosis (n = 1091)	5.3 ± 1.8	140 (13)	3.1 ± 1.8	681 (62)	69 (6.3)
Cardioembolism (n = 618)	5.5 ± 1.9	111 (18)	3.2 ± 1.8	400 (65)	52 (8.4)
Other determined etiology (n = 68)	4.8 ± 1.6	4 (6)	2.5 ± 1.9	29 (43)	6 (7.4)
Undetermined etiology (n = 108)	5.2 ± 1.7	11 (10)	2.5 ± 1.9	51 (47)	5 (4.6)
*p*-Value	<0.001	<0.001	<0.001	<0.001	<0.001

Data are expressed as mean ± standard deviation or n (%); ANOVA or chi-square test; mRS, modified Rankin Scale; TOAST, Trial of ORG 10172 in Acute Stroke Treatment.

**Table 5 biomedicines-10-02185-t005:** Correlation of serum uric acid level and hyperuricemia with measured variables and outcomes in 3370 patients with acute ischemic stroke.

Uric Acid Level ^a^									
Variables	All Patients (*n* = 3370)			Men (*n* = 1925)			Women (*n* = 1945)		
	Coefficient	R^2^	*p*-Value	Coefficient	R^2^	*p*-Value	Coefficient	R^2^	*p*-Value
Age	0.019	<0.001	0.889	−0.349	0.002	0.060	1.111	0.022	<0.001
Body mass index	0.408	0.027	<0.001	0.345	0.019	<0.001	0.429	0.027	<0.001
Hemoglobin	0.107	0.008	<0.001	0.087	0.005	0.001	−0.021	<0.001	0.469
White blood cells	0.151	0.009	<0.001	0.160	0.010	<0.001	0.129	0.006	0.003
Glucose	−1.967	0.002	0.016	−3.879	0.007	<0.001	0.469	<0.001	0.716
Creatinine	0.124	0.035	<0.001	0.100	0.022	<0.001	0.138	0.045	<0.001
Cholesterol	2.063	0.007	<0.001	3.977	0.025	<0.001	0.752	<0.001	0.273
Triglyceride	8.17	0.022	<0.001	7.157	0.013	<0.001	8.932	0.038	<0.001
Admission NIHSS score	0.131	0.001	0.069	0.278	0.005	0.002	0.191	0.002	0.107
Discharge NIHSS score	0.239	0.002	0.012	0.206	0.002	0.089	0.528	0.008	<0.001
Discharge BI score	−0.024	<0.001	0.947	−0.179	<0.001	0.702	−1.229	0.003	0.030
Discharge mRS score	−0.0003	<0.001	0.988	−0.011	<0.001	0.653	0.074	0.005	0.005
**Hyperuricemia (n = 400) ^b^**									
**Variables**	**All 3370 Patients**			**Men (*n* = 1925)**			**Women (*n* = 1945)**		
	**Yes**	**No**	***p* Value**	**Yes**	**No**	***p* Value**	**Yes**	**No**	***p* Value**
Hypertension (n = 2363)	316 (13)	84 (8)	<0.001	140 (10)	51 (9)	0.188	176 (17)	33 (8)	<0.001
Diabetes mellitus (n = 1157)	152 (13)	248 (11)	0.102	59 (9)	132 (10)	0.376	93 (18)	116 (12)	0.002
Dyslipidemia (n = 718)	101 (14)	299 (11)	0.044	55 (14)	136 (9)	0.007	46 (15)	163 (14)	0.928
Heart disease (n = 1085)	173 (16)	227 (10)	<0.001	71 (12)	120 (9)	0.019	102 (20)	107 (11)	<0.001
Prior stroke (n = 769)	90 (12)	310 (12)	0.871	43 (9)	148 (10)	0.594	47 (15)	162 (14)	0.716
Current smoker (n = 717)	73 (10)	327 (12)	0.119	67 (10)	124 (10)	0.873	6 (9)	203 (15)	0.279
Alcohol consumption (n = 227)	20 (9)	380 (12)	0.166	20 (9)	171 (10)	0.721	0 (0)	209 (15)	0.603
History of cancer (n = 220)	20 (9)	380 (12)	0.235	8 (8)	183 (10)	0.504	12 (11)	197 (15)	0.266
Uremia (n = 67)	12 (18)	388 (12)	0.127	6 918)	185 (10)	0.141	6 (18)	203 (14)	0.462
Complications (n = 373)	64 (17)	336 (11)	0.002	22 (13)	169 (10)	0.229	42 (21)	167 (13)	0.007
Discharge mRS > 2 (n = 1658)	234 (14)	166 (10)	<0.001	90 (11)	101 (9)	0.189	144 (17)	65 (11)	<0.001
Death (n = 133)	34 (26)	366 (11)	<0.001	12 (18)	179 (10)	0.038	22 (34)	187 (14)	<0.001

^a^ linear regression test; ^b^ Fisher’s exact test; BI, Barthel index; mRS, modified Rankin Scale; NIHSS, National Institutes of Health Stroke Scale.

**Table 6 biomedicines-10-02185-t006:** Comparison of measured variables with different quartiles of uric acid.

	Quartile 1 (<4.1) ^a^	Quartile 2 (4.1–5.1) ^a^	Quartile 3 (5.2–6.2) ^a^	Quartile 4 (>6.2) ^a^	*p*-Value
All patients (*n* = 3370)					
Number of patients	889	847	811	823	
Age (years)	71.0 ± 13.6	70.3 ± 13.5	70.4 ± 13.3	70.8 ± 14.1	0.619
Female gender	510 (57)	386 (46)	280 (35)	269 (33)	<0.001
NIHSS score on admission	7.5 ± 7.2	6.4 ± 6.9	6.3 ± 6.6	7.5 ± 7.9	<0.001
mRS score >2 at discharge	510 (57)	386 (46)	280 (45)	403 (49)	<0.001
Death at discharge	40 (4.5)	26 (3.1)	19 (2.3)	48 (5.8)	0.001
Male patients (*n* = 1925)					
Number of patients	379	461	531	554	
Age (years)	69.3 ± 13.9	68.0 ± 12.9	68.0 ± 13.2	67.4 ± 13.9	0.208
NIHSS score on admission	6.1 ± 6.1	5.6 ± 6.2	5.8 ± 6.0	6.6 ± 7.3	0.080
mRS score >2 at discharge	195 (51)	185 (40)	211 (40)	227 (41)	0.001
Death at discharge	22 (5.8)	13 (2.8)	8 (1.5)	25 (4.5)	0.002
Female patients (*n* = 1445)					
Number of patients	510	386	280	269	
Age (years)	72.3 ± 13.2	73.0 ± 13.7	74.9 ± 12.1	77.8 ± 11.6	<0.001
NIHSS score on admission	8.5 ± 7.7	7.2 ± 7.4	7.4 ± 7.6	9.3 ± 8.7	0.002
mRS score >2 at discharge	308 (60)	202 (52)	150 (54)	180 (67)	<0.001
Death at discharge	18 (3.5)	13 (3.4)	11 (3.9)	23 (8.6)	0.005

Data are expressed as mean ± standard deviation or n (%); ^a^ range of uric acid level in each quartile are derived from all patients (mg/dL); mRS, modified Rankin Scale; NIHSS, National Institutes of Health Stroke Scale.

**Table 7 biomedicines-10-02185-t007:** Multivariable analysis of factors influencing unfavorable outcomes (modified Rankin Scale score > 2) in 3370 patients with acute ischemic stroke.

Characteristics	All Patients (*n* = 3370)			Male Patients (*n* = 1925)		Female Patients (*n* = 1445)		
	OR (95% CI) ^b^	Model I	Model II	OR (95% CI)	Model I	OR (95% CI) ^b^	Model I	Model II
		*p* Value	*p* Value		*p* Value		*p* Value	*p* Value
Age	1.051 (1.042–1.061)	<0.001	<0.001	1.045 (1.033–1.057)	<0.001	1.055 (1.040–1.070)	<0.001	<0.001
Admission NIHSS score	1.422 (1.377–1.467)	<0.001	<0.001	1.434 (1.377–1.494)	<0.001	1.409 (1.342–1.479)	<0.001	<0.001
Female gender	1.200 (0.955–1.508)	0.110	0.117	-	-	-	-	-
Hemoglobin	0.952(0.899–1.008)	0.147	0.094	0.926 (0.861–0.996)	0.038	0.992 (0.908–1.084)	0.862	0.815
Platelet	0.999 (0.998–1.001)	0.424	0.436	1.000 (0.998–1.002)	0.736	0.999 (0.997–1.001)	0.450	0.450
White blood cells	1.058 (1.016–1.102)	0.005	0.007	-	-	1.131 (1.060–1.207)	<0.001	<0.001
Glucose	1.002 (1.001–1.004)	0.012	0.005	-	-	1.003 (1.001–1.005)	0.043	0.035
Creatinine	0.978 (0.862–1.110)	0.929	0.732	0.900 (0.771–1.049)	0.178	1.074 (0.927–1.243)	0.342	0.432
Cholesterol	1.003 (1.001–1.006)	0.014	0.019	1.004 (1.000–1.008)	0.016	1.003 (0.999–1.007)	0.136	0.131
Triglyceride	1.000 (0.999–1.001)	0.509	0.662	1.001 (0.999–1.002)	0.337	1.000 (0.997–1.002)	0.661	0.527
Hypertension	1.024 (0.825–1.270)	0.657	0.832	-	-	1.135 (0.802–1.607)	0.474	0.592
Diabetes mellitus	1.147 (0.904–1.455)	0.309	0.259	1.389 (1.075–1.796)	0.012	1.019 (0.692–1.501)	0.925	0.848
Dyslipidemia	0.837 (0.653–1.072)	0.218	0.159	0.952 (0.691–1.312)	0.763	0.781 (0.533–1.144)	0.204	0.166
Heart disease	0.888 (0.711–1.110)	0.428	0.298	0.997(0.753–1.320)	0.981	0.848 (0.596–1.208)	0.361	0.290
Prior stroke	1.574 (1.254–1.977)	<0.001	<0.001	1.683 (1.276–2.220)	<0.001	1.507 (1.033–2.197)	0.033	0.032
Current smoker	1.023 (0.787–1.328)	0.838	0.867	1.050 (0.810–1.361)	0.713	0.629 (0.293–1.355)	0.236	0.212
Alcohol consumption	0.926 (0.618–1.388)	0.688	0.711	-	-	-	-	-
History of cancer	1.655 (1.139–2.405)	0.011	0.008	-	-	1.920 (1.102–3.344)	0.021	0.017
Uremia	1.492 (0.557–3.994)	0.670	0.426	2.525 (0.730–8.727)	0.143	-	-	-
Quartile 1 of uric acid ^a^	1.359 (1.068–1.730)	0.013	-	1.561 (1.144–2.130)	0.005	1.282 (0.882–1.863)	0.192	-
Quartile 4 of uric acid ^a^	0.972 (0.766–1.233)	0.815	-	0.915 (0.682–1.228)	0.554	1.019 (0.695–1.493)	0.925	-
Hyperuricemia	1.054 (0.764–1.452)	-	0.749	-	-	1.115 (0.702–1.772)	-	0.644

Model I: including uric acid quartiles 1 to 4 derived from all patients, ranges of Quartiles 1 and 4 are <4.1 mg/dL and >6.2 mg/dL, respectively; Model II: replacing uric acid quartiles with hyperuricemia; ^a^ Using Quartiles 2 and 3 of uric acid as reference percentiles; ^b^ Representing results in Model I; CI, confidence interval; NIHSS, National Institutes of Health Stroke Scale; OR, odds ratio.

**Table 8 biomedicines-10-02185-t008:** Multivariable analysis of factors influencing death in 3370 patients with acute ischemic stroke.

Characteristics	All Patients (*n* = 3370)				Male Patients (*n* = 1925)			Female Patients (*n* = 1445)			
	OR (95% CI) ^b^	Model I	Model II	Model III	OR (95% CI) ^b^	Model I	Model II	OR (95% CI) ^b^	Model I	Model II	Model III
		*p* Value	*p* Value	*p* Value		*p* Value	*p* Value		*p* Value	*p* Value	*p* Value
Age	1.019 (1.001–1.036)	0.025	0.036	0.030	1.023 (1.000–1.046)	0.047	0.053	1.022 (0.994–1.052)	0.095	0.126	0.134
Admission NIHSS score	1.134 (1.111–1.157)	<0.001	<0.001	<0.001	1.166 (1.131–1.202)	<0.001	<0.001	1.124 (1.090–1.159)	<0.001	<0.001	<0.001
Hemoglobin	0.952 (0.862–1.053)	0.360	0.339	0.291	0.887 (0.772–1.020)	0.092	0.062	-	-	-	-
Platelet	-	-	-	-	1.001 (0.998–1.004)	0.568	0.431	-	-	-	-
White blood cells	1.110 (1.049–1.176)	<0.001	<0.001	<0.001	1.137 (1.041–1.243)	0.005	0.005	1.085 (1.002–1.174)	0.033	0.044	0.044
Glucose	1.002 (1.000–1.004)	0.078	0.084	0.061	-	-	-	1.001 (0.998–1.004)	0.466	0.569	0.541
Creatinine	1.064 (0.883–1.283)	0.472	0.515	0.586	-	-	-	1.212 (1.033–1.422)	0.019	0.018	0.029
Triglyceride	-	-	-	-	1.001 (0.997–1.004)	0.682	0.920	-	-	-	-
Diabetes mellitus	-	-	-	-	-	-	-	1.477 (0.795–2.743)	0.201	0.217	0.210
Heart disease	1.844 (1.205–2.820)	0.004	0.005	0.004	1.265 (0.688–2.328)	0.449	0.542	2.394 (1.253–4.578)	0.007	0.008	0.008
History of cancer	2.100 (1.114–3.958)	0.033	0.022	0.022	-	-	-	2.136 (0.934–4.885)	0.070	0.072	0.067
Uremia	1.410 (0.376–5.295)	0.669	0.611	0.571	-	-	-	-	-	-	-
Quartile 1 of uric acid ^a^	1.474 (0.895–2.427)	0.127	-	-	3.109 (1.518–6.367)	0.002	-	0.692 (0.328–1.458)	0.333	-	-
Quartile 4 of uric acid ^a^	1.498 (0.942–2.381)	0.088	-	-	1.509 (0.750–3.036)	0.249	-	1.215 (0.640–2.304)	0.552	-	-
Hyperuricemia	1.518 (0.935–2.463)	-	0.091	-	1.215 (0.554–2.665)	-	0.627	1.905 (1.009–3.596)	-	0.047	-
Uric acid	1.111 (0.973–1.169)	-	-	0.171	-	-	-	1.151 (1.014–1.307)	-	-	0.029

Model I: including uric acid quartiles 1 to 4, Quartiles 1 and 4 for all patients, male and female patients: <4.1 mg/dL and >6.2 mg/dL, respectively; Model II: replacing uric acid quartiles with hyperuricemia; ^a^ Using Quartiles 2 and 3 of uric acid as reference percentiles; ^b^ Representing results in Model I; CI, confidence interval; NIHSS, National Institutes of Health Stroke Scale; OR, odds ratio.

## Data Availability

The data presented in this study are available on request from the corresponding author.

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
