# Peer review of "J-Shaped Relationship of Serum Uric Acid with Unfavorable Short-Term Outcomes among Patients with Acute Ischemic Stroke"

_biomedicines, 2022, doi:10.3390/biomedicines10092185_

Round 1

Reviewer 1 Report

This clinical study reported the relationship between serum uric acid and unfavorable short-term outcome in stroke patients. The data were collected and analyzed from 3370 patients with acute ischemic stroke. They concluded that lower uric acid levels are a predictor for unfavorable outcomes and death in men, and higher uric acid levels are a prdictor for death in women. 

There was no difference in uric acid levels present in Table 2 when the two groups were compared in both men and women. The average uric acid levels were among the normal ranges (4.6-7.6 mg/dl in men and 2.3-6.6 mg/dl in women). 

The authors did have positive results when the uric acid levels were divided into 4 quartiles (< 4.1 mg/dl, 4.1-5.1 mg/dl, 5.2-6.2 mg/dl, and > 6.2 mg/dl). However, the normal range of uric acid levels differs in men and women. Should the data from men and women be analyzed using the same quartiles? 

Hyperuricemia, unfavorable outcomes, and death were observed in 12%, 49%, and 4% of all patients, respectively. The scale of uric acid levels in the figures is 0-18 mg/dl.  Is it possible to analyze the relationship between the uric acid levels and unfavorable outcomes or death in these patients with hyperuricemia?

What mechanism supports the conclusion that a lower uric acid level is a predictor of unfavorable outcomes or death in men?  This conclusion should be supported by solid evidence.

Author Response

Response to Reviewer 1 Comments

Manuscript ID: Biomedicines-1877652

Title: J-shaped relationship of serum uric acid with unfavorable short-term outcomes among patients with acute ischemic stroke

Thanks to reviewer’s precious comments. We have checked the manuscript and have made essential revisions according to reviewer’s comments point-by-point.

Point 1: There was no difference in uric acid levels present in Table 2 when the two groups were compared in both men and women. The average uric acid levels were among the normal ranges (4.6-7.6 mg/dl in men and 2.3-6.6 mg/dl in women).

Response: Compared with other parameters, no significant difference of the uric acid level was observed between patients with favorable and unfavorable outcomes. That was because both higher and lower levels of uric acid were associated with unfavorable outcomes. Such correlation could not be identified by univariate analysis such as unpaired t or Mann-Whitney U test, but was demonstrated successfully by using the least squares regression analyses and the quartile distribution of uric acid.

Point 2: The authors did have positive results when the uric acid levels were divided into 4 quartiles (< 4.1 mg/dl, 4.1-5.1 mg/dl, 5.2-6.2 mg/dl, and > 6.2 mg/dl). However, the normal range of uric acid levels differs in men and women. Should the data from men and women be analyzed using the same quartiles?

Response: Thanks for reviewer’s important remind. We have analyzed two different patterns of the quartile distribution of uric acid in Tables 6-8. The first method was using the same quartiles derived from all patients for all patients, men and women. The second method was using different quartiles derived from men and women separately. All the statistical results were similar by using either the first or the second method, except for the number of patients categorized in different quartiles. To simplify the clinical application, we prefer to use the first method with the same quartile distributions for both men and women. The results of the second method have been demonstrated in supplementary Tables (Table S1-3) corresponding to Tables 6-8. We have to apologize for the errors in the original version of the manuscript that data in Tables 6-8 were using the same quartile distributions derived from all patients. We have corrected the errors and made essential revisions for Tables 6-8.

Point 3: Hyperuricemia, unfavorable outcomes, and death were observed in 12%, 49%, and 4% of all patients, respectively. The scale of uric acid levels in the figures is 0-18 mg/dl. Is it possible to analyze the relationship between the uric acid levels and unfavorable outcomes or death in these patients with hyperuricemia?

Response: We have conducted additional Mann-Whitney U test to analyze the relationship between the uric acid levels and unfavorable outcomes or death in patients with hyperuricemia, and have presented the results in Line 148-153.

Point 4: What mechanism supports the conclusion that a lower uric acid level is a predictor of unfavorable outcomes or death in men? This conclusion should be supported by solid evidence.

Response: We have added a paragraph in Discussion section to describe the mechanism of a lower uric acid level predicting unfavorable outcomes or death in Line 358-371.

Reviewer 2 Report

Dear editor,

I liked to review the manuscript detailed below submitted to biomedicine.

J-shaped relationship of serum uric acid with unfavorable short-term outcomes among patients with acute ischemic stroke+

The authors investigated the relevance for the short-term outcome in stroke patients related to the detected levels for uric acid. In total 3,370 patients were studied. An unfavorable outcome and death (modified Rankin scale >2) was found in males with lower uric acid levels, while higher levels for uric acid was identified as predictor for death in women.

I would recommend accepting the paper; it contains important information for the clinical practice.     

However, I have some points of criticism, which potentially could improve the quality of the manuscript:

1.       I would prefer to use direct range values for the uric acid levels instead mentioning second or third quartile. There is no clue what that means when only reading the abstract.

2.       I would avoid propaedeutic contents in the introduction, such as stroke is a leading cause of death…. . This is well known, and it do not add any additional information to the manuscript. I would extend a little the second part of the introduction, and delete the first one entirely.

3.       The results are presented very well. I like it! Especially the graphs. Here is also important to point out the ranges. Using only the term quartile one or two is difficult. Figures should be readable alone, without searching for explanation in the text.

4.       The discussion is well balanced and it points out the results well. The first sentence seem to be a little misplaced. “The role of uric acid in AIS is complicated”, is this a joke? I would delete it.

5.       Please add any sentences with information regarding implication for the clinical routine. How this could be implemented in the clinical practice.          

Author Response

Response to Reviewer 2 Comments

Manuscript ID: Biomedicines-1877652

Title: J-shaped relationship of serum uric acid with unfavorable short-term outcomes among patients with acute ischemic stroke

Thanks to reviewer’s precious comments. We have checked the manuscript and have made essential revisions according to reviewer’s comments point-by-point.

Point 1: I would prefer to use direct range values for the uric acid levels instead mentioning second or third quartile. There is no clue what that means when only reading the abstract.

Response: We have added the range of each quartile of uric acid in the Abstract section.

Point 2: I would avoid propaedeutic contents in the introduction, such as stroke is a leading cause of death…. . This is well known, and it do not add any additional information to the manuscript. I would extend a little the second part of the introduction, and delete the first one entirely

Response: We have deleted unnecessary statements and reorganized the contents of the Introduction section with further information about uric acid.

Point 3: The results are presented very well. I like it! Especially the graphs. Here is also important to point out the ranges. Using only the term quartile one or two is difficult. Figures should be readable alone, without searching for explanation in the text.

Response: We have added the range of uric acid level of each quartile in all corresponding Tables and Figures.

Point 4: The discussion is well balanced and it points out the results well. The first sentence seem to be a little misplaced. “The role of uric acid in AIS is complicated”, is this a joke? I would delete it.

Response: We have deleted “The role of uric acid in AIS is complicated”.

Point 5: Please add any sentences with information regarding implication for the clinical routine. How this could be implemented in the clinical practice.

Response: We have added further information regarding clinical application of uric acid in evaluation of stroke outcomes in Discussion section Line 387-392.

Round 2

Reviewer 1 Report

There are two sets of A, B, and C in Figure 3. Are they duplicate? Please confirm.

Author Response

Response to Reviewer 1 Comments

Manuscript ID: Biomedicines-1877652

Title: J-shaped relationship of serum uric acid with unfavorable short-term outcomes among patients with acute ischemic stroke

Thanks to reviewer’s precious comments. We have checked the manuscript and have made essential revisions according to reviewer’s comments point-by-point.

Comments and Suggestions for Authors

There are two sets of A, B, and C in Figure 3. Are they duplicate? Please confirm.

Response: In V2 version of the manuscript, we deleted old Figure 3 and replaced a new Figure 3 with different items of quartile distribution (Quartile 1 - 4). We have rechecked Figure 3 carefully but failed to find two sets of A, B, C in Figure 3. However, we would like to thank to reviewer’s remind because we found errors in the legends of Figure 2 and have corrected it in V3 version.